# AMPK-Mediated Phosphorylation of Nrf2 at S374/S408/S433 Favors Its βTrCP2-Mediated Degradation in KEAP1-Deficient Cells

**DOI:** 10.3390/antiox12081586

**Published:** 2023-08-09

**Authors:** Eleni Petsouki, Sylvia Ender, Shara Natalia Sosa Cabrera, Elke H. Heiss

**Affiliations:** 1Department of Pharmaceutical Sciences, University of Vienna, 1090 Vienna, Austria; a01406211@unet.univie.ac.at (S.E.); shara.natalia.sosa.cabrera@univie.ac.at (S.N.S.C.); elke.heiss@univie.ac.at (E.H.H.); 2Vienna Doctoral School of Pharmaceutical, Nutritional and Sport Sciences, University of Vienna, 1090 Vienna, Austria

**Keywords:** Nrf2, AMPK, βTrCP, E3 ubiquitin ligase, transcription factor

## Abstract

Nrf2 is a transcription factor facilitating cells’ resilience against redox and various other forms of stress. In the absence of stressors, KEAP1 and/or βTrCP mediate the ubiquitination of Nrf2 and prevent Nrf2-dependent gene expression and detoxification. AMPK regulates cellular energy homeostasis and redox balance. Previous studies indicated a potential Nrf2-AMPK cooperativity. In line with this, our lab had previously identified three AMPK-dependent phosphorylation sites (S374/408/433) in Nrf2. Given their localization in or near the Neh6 domain, known to regulate βTrCP-mediated degradation, we examined whether they may influence the βTrCP-driven degradation of Nrf2. By employing expression plasmids for WT and triple mutant (TM)-Nrf2 (Nrf2^S374/408/433→A^), (co)immunoprecipitation, proximity ligation, protein half-life, knockdown, ubiquitination experiments, and qPCR in *Keap1*-null mouse embryonic fibroblasts, we show that TM-Nrf2^S→A374/408/433^ had enhanced stability due to impeded interaction with βTrCP2 and reduced ubiquitination in comparison to WT-Nrf2. In addition, TM-Nrf2 elicited higher expression of the Nrf2 target gene *Gclc*, potentiated in the presence of a pharmacological AMPK activator. Overall, we propose that AMPK-dependent phospho-sites of Nrf2 can favor its βTrCP2-mediated degradation and dampen the extent of Nrf2 target gene expression. Therefore, targeting AMPK might be able to diminish Nrf2-mediated responses in cells with overactive Nrf2 due to KEAP1 deficiency.

## 1. Introduction

Nrf2 (Nuclear factor E2 p45-related factor 2) is a ubiquitously expressed leucine zipper transcription factor [1,2,3] that plays a critical role in defending the cell against oxidative stress, promoting detoxification, regulating inflammation, and supporting cellular resilience [4,5,6]. Nrf2 consists of different domains, including a KEAP1-binding domain (Neh2), transactivation domains (Neh3–5), a RXRα binding domain (Neh7), a βTrCP binding domain (Neh6), and the DNA binding domain (Neh1) [2]. Nrf2 activity is predominantly regulated via its abundance on a (a) transcriptional, (b) posttranscriptional, (c) epigenetic level, or (d) via ubiquitination and proteasomal degradation [7].

The major Nrf2 ubiquitination pathways include the canonical KEAP1 (Kelch-like-ECH-associated protein)-dependent and the non-canonical βTrCP (β-transducin-repeat containing protein)-mediated pathway, which usually can be pictured as a limiter and regulator valve for Nrf2 signaling, respectively [8,9,10,11]. In the absence of stressors, the redox-regulated KEAP1, an adaptor for Cul3-dependent ubiquitin ligases, interacts with the Neh2 domain of Nrf2 in a 2:1 ratio and mediates the predominant share of proteasomal degradation of Nrf2. Alternatively, βTrCP, as part of SCF ubiquitin–protein ligase complexes, can interact with the Neh6 domain of Nrf2 and also lead to the ubiquitination and degradation of the transcription factor [11,12,13,14,15,16,17,18]. In the presence of stress signals, ubiquitination of Nrf2 ceases, allowing Nrf2 to migrate into the nucleus and activate over 250 Nrf2 target genes with an ARE (antioxidant response element) sequence in their regulatory regions, thus influencing redox homeostasis, detoxification, inflammation, and metabolism [19,20,21]. Given the fundamental role of Nrf2 in cellular homeostasis, its dysregulation in the form of either too much or too little activity can lead to or aggravate diseases (inflammatory and metabolic-related diseases, aging, and cancer) [6,10,21,22,23,24,25,26,27,28].

For instance, mutations in KEAP1 can cause this dysregulation, leading to the constitutive activation of NRF2 signaling [29]. KEAP1 mutations are present in various pathological conditions, including neurological and metabolic disorders, cardiovascular diseases, and cancer [8,28,30,31,32,33], with a high prevalence in lung and pancreatic cancer [34,35]. These cancers then show hijacked NRF2 hyperactivation, which facilitates high tumor growth, therapy resistance, and metastasis [33,36,37,38] due to persistent antioxidant [39] and detoxification pathways [40], allowing cancer cells to fight stress induced by aberrant proliferation and therapy [38,41].

AMP-activated protein kinase (AMPK) is a serine/threonine kinase that is formed as a heterotrimeric complex by a catalytic α subunit and regulatory β and γ subunits [42]. AMPK activity is susceptible to regulation by different cues, including phosphorylation, inhibition of dephosphorylation at Thr172, and by allosteric modulation [43]. Its structural organization renders AMPK a sensor to changes in AMP/ATP and ADT/ATP ratios, and it activates signaling pathways resulting in cellular energy homeostasis [44]. Taking into consideration that energy metabolism affects cellular behavior, AMPK was also reported to influence proliferation, cell growth, autophagy, inflammation, and redox balance [45,46,47,48,49,50,51]. Therefore, its dysregulation can be observed in many diseases and pathological conditions such as diabetes, obesity, inflammation, cancer, and redox stress [52,53,54,55,56].

Concomitant activation by a plethora of exogenous insults and initiation of overlapping stress-relieving responses [57,58] quite early indicated potential cooperativity between Nrf2 and AMPK signaling. Meanwhile, their mutual multimodal crosstalk could be shown to occur among others via direct phosphorylation of Nrf2 at different serine residues or indirectly via the AMPK/GSK-beta/TrCP axis, the p62/KEAP1/autophagy axis, or epigenetic mechanisms [22]. In most cases, AMPK activation turned out to regulate the Nrf2-dependent response in a positive manner [11,59,60,61,62,63,64].

In one of our previous studies, we could identify AMPK-dependent phosphorylation sites at S374/408/433 in Nrf2. Their mutation to alanine reduced the expression of selected Nrf2 target genes in an AMPK-dependent manner [65]. These residues are located in and near the Neh6 domain of Nrf2, hosting the two β-TrCP binding sites at 343-DSGIS-347 and 382-DSAPGS-387 [25], which are implicated in Nrf2 degradation via the proteasome [11]. Given the existence of (a) AMPK-dependent phospho-sites in Nrf2, (b) their localization near the Neh6 domain, and (c) somatic mutations in the *KEAP1* leading to an anomalous activation of Nrf2, we set out to investigate the effect of the three AMPK-dependent phosphorylation sites of Nrf2 for their role in its βTrCP-driven degradation in a KEAP1 depleted background. Delving into the interplay between AMPK and Nrf2 would provide valuable and novel insights into the network of cellular stress resilience mechanisms. It might also pave the way for the rational use of pharmacological AMPK activators in order to modulate Nrf2-mediated responses, especially in cases of aberrant Nrf2 signaling [66] pertinent to certain disease contexts caused by KEAP1 deficiency.

## 2. Materials and Methods

### 2.1. Cell Culture

KEAP1-depleted mouse embryonic fibroblasts (MEFs*^Keap1−/−^*) were kind gifts from Thomas Kensler (University of Pittsburgh, Pittsburgh, PA, USA) [67]. Media, serum, and supplements for cell culture were purchased from Sigma-Aldrich, Biowest, and Corning, respectively. MEFS were maintained in Dulbecco’s Modified Eagle’s Medium (DMEM) plus 10% heat-inactivated fetal bovine serum and 1% Penicillin/Streptomycin/L-Glutamine (P/S/G, Corning Product, Corning, NY, USA). OPTI-MEM I (1×) (Gibco) was used for transient transfections.

### 2.2. Reagents and Compound Treatment

Cycloheximide (CHX) was purchased from Sigma-Aldrich (North Brunswick, NJ, USA), A769662 came from ApexBio (Houston, TX, USA), and MG132 was purchased from ChemCruz (Huissen, The Netherlands). The working concentration of (a) CHX is 100 μM, (b) A769662 is 50 μM, and (c) MG132 is 10 μΜ.

### 2.3. Plasmids

The mammalian expression vectors pcDNA3-EGFP-C4-Nrf2 (EGFP-Myc-WT-Nrf2) #21549, pGLUE-HA-FBXW11#36969, and His-Ubiquitin (#31815) were obtained from Addgene and pcDNA3.1 vector from Invitrogen. The triple serine–alanine mutant form of Nrf2 (EGFP-MycTM-Nrf2) was designed as previously described [65].

### 2.4. siRNA Sequences

Mouse *Fbxw11*-specific smart pool siRNA was obtained from Dharmacon (#L-058886-00-0005, Lafayette, CO, USA). Non-targeting siRNA pool was obtained from Merck (#SIC007, Rahway, NJ, USA). The working concentration was 100 nM.

### 2.5. Transient Transfections

Transfections with plasmids in MEFs*^Keap1−/−^* cells were performed by using Lipofectamine LTX with PLUS^TM^ Reagent (Invitrogen, Bridgewater, NJ, USA) according to the manufacturer’s instructions. Transfection with siRNA or co-transfections with siRNA and plasmids in MEFs*^Keap1−/−^* cells were performed by using Lipofectamine 3000 (Invitrogen) according to the manufacturer’s instructions.

### 2.6. Immunoblotting

MEFs*^Keap1−/−^* cells were lysed as previously described [68]. Cell lysates were separated by SDS-PAGE, and proteins were blotted to PVDF membranes. For detection, we used antibodies against the tags of the transfected proteins of interest in order to avoid the interference of endogenous proteins and the poor performance of commercially available antibodies for our purpose. Antibodies were used against Myc-Tag (#2276S, 1:1000 dilution) from Cell Signaling (Danvers, MA, USA), HA-Tag (#3724S, 1:1000 dilution) from Cell Signaling, beta-actin (#8457S, 1:1000 dilution) from Cell Signaling and FBXW11 (#13149-1-AP, 1:000 dilution) from Proteintech (Rosemont, IL, USA). Densitometric quantification was performed with ImageQuant™ TL, Cytiva (Tokyo, Japan)

### 2.7. Immunoprecipitation

MEFs*^Keap1−/−^* cells were transfected with the indicated constructs for 48 h, and then treated with MG132 proteasomal inhibitor 4 h before their lysis. Cells were lysed and further processed as previously described [65].

### 2.8. Proximity Ligation Assay (PLA) and Indirect Immunofluorescence

MEFs*^Keap1−/−^* cells were grown on 0.1% gelatin-coated glass coverslips, transfected with the indicated constructs for 48 h, and then treated with MG132 proteasomal inhibitor 4 h before their fixation with 4% Paraformaldehyde (#j19943-k2, Thermo Fischer Scientific, Vienna, Austria). After the fixation, they were permeabilized in 0.1% Triton X and blocked in 1× PLA blocking solution for 1 h at 37 °C in a humidity chamber. Then, they were subjected to PLA (#DUO92004, DUO92002, Merck). Fixed samples were incubated with HA and MYC-Tag primary antibodies (1:1000 dilution) raised in separate species (mouse and rabbit) that bind to HA-βTrCP2 and Myc-Nrf2 forms, respectively in a humidity chamber at 4 °C overnight. Then, the samples were washed twice with Duolink^®^ Wash Buffer A for 5 min. Secondary antibodies conjugated with oligonucleotides (PLA probe MINUS and PLA probe PLUS) were diluted with the Duolink^®^ Antibody Diluent at 1:5 PLA probe dilution and incubated in a humidity chamber at 37 °C for 1 h. The next step included the ligation and amplification process according to the manufacturer’s instructions: The DNA oligonucleotides hybridized with the two PLA probes, and the enzyme formed a ligated (closed) circle when the probes were in close proximity. Finally, one of the PLA probes acted as a primer for the polymerase, generating a repeating, concatemeric product that was still tethered to the secondary antibody. Labeled oligonucleotides were then hybridized with the amplified DNA to yield a signal that was easily detectable by fluorescence microscopy. Samples were mounted with Diamond DAPI (#P36966, Invitrogen) to counterstain the nuclei. *Z*-stacks were obtained with a Leica DMi8 inverted confocal microscope, and images were analyzed by LAS X (3.1.2) software.

### 2.9. Quantitative Chain Polymerase Reaction (qPCR)

RNA isolation, reverse transcription, and qPCR were performed as previously described [69]. mRNA levels of the genes of interest were normalized to the indicated control via the ΔΔCt method. Primers used for qPCR were *Hmox*1 Fw:5′-AAGCCGAGAATGCTGAGTTCA-3′, *Hmox1* Rv:5′-GCCGTGTAGATATGGTACAAGGA, *Gclc* Fw:5′-TGGCCACTATCTGCCCAATT-3′, *Gclc* Rv:5′-GTCTGACACGTAGCCTCGGTAA-3′, *TBP* Fw:5′-TCTACCGTGAATCTTGGCTGT-3′, and *TBP* Rv:5′-GTCCGTGGCTCTCTTATTCTCA-3′.

### 2.10. In Vitro Ubiquitinylation Assay

MEFs*^Keap1−/−^* cells were transiently co-transfected with HA-βTrCP2, His ubiquitin, and EGFP-Myc-WT-Nrf2 or EGFP-Myc-TM-Nrf2 for 48 h and further processed to the in vitro ubiquitination assay as previously described [70].

### 2.11. Statistical Analysis

The statistical analyses were performed by Software GraphPad Prism 9. Student’s *t*-test or one-way ANOVA was performed for the quantification analysis of the results based on the number of comparison groups, followed by the proper ad hoc test. Data are presented as means ± SEM from at least three independent biological replicates. Regarding the statistical analysis performed using multiple comparison one-way ANOVA, it was followed with Sidak’s correction, * *p* < 0.05 and ** *p* < 0.01.

## 3. Results

### 3.1. Mutation of the AMPK-Dependent Phospho-Sites S374, S408, and S433 to Alanine Stabilizes Nrf2 in Keap1^−/−^ Cells in a βTrCP2-Dependent Manner

The main degradation pathways of Nrf2 are (a) the KEAP1-dependent pathway and (b) the KEAP1-independent pathways, involving the βTrCP E3 ubiquitin ligase subunit of the SCF^βΤrCP^ complex. To address whether the previously identified phospho-sites of Nrf2 have a role in the stability of the protein in the context of βTrCP-mediated degradation, we used KEAP1-depleted mouse embryonic fibroblasts (MEFs) to overcome KEAP1 interference. Cells were transfected with a wild form of Nrf2 (GFP-MYC-WT-Nrf2) or a triple mutated version of Nrf2 (GFP-MYC-TM-NRF2) in which S374, S408, and S433 were mutated to alanine. To monitor the time-dependent decay of Nrf2 protein, de novo protein synthesis was blocked by treatment with cycloheximide. The triple mutated form of Nrf2 had a higher half-life in comparison to the wild-type form (~18 min vs. 7 min) (Figure 1A,B). This finding showed that mutation of the identified phospho-sites renders the protein more stable and less susceptible to degradation. To further investigate whether the increased stability of TM-Nrf2 was indeed related to βTrCP-mediated degradation, we knocked down βTrCP2 by siRNA and repeated the experimental setup from above. Downregulation of βTrCP2 resulted in similar kinetics of degradation between the WT-Nrf2 and the TM-Nrf2 (Figure 1C,D). These data suggest that the identified residues control the abundance of Nrf2 in a βTrCP2-dependent manner. Successful depletion of endogenous mouse βTrCP2 by *Fbxw11* siRNA was confirmed by immunoblot in transfected MEFs*^Keap1−^*^/*−*^ (Appendix A).

### 3.2. Mutation of the AMPK-Dependent Phospho-Sites to Alanine Impedes the Interaction of Nrf2 with β-TrCP2 in Keap1^−/−^ Cells

To further corroborate the role of phospho-S374, -S408, and -S433 in βTrCP2-mediated degradation of Nrf2, we tested the binding of βTrCP2 to Nrf2. We addressed this question by employing a co-immunoprecipitation and proximity ligation assay (PLA) in *Keap1^−^*^/*−*^ cells transfected with GFP-Myc tagged WT or TM-Nrf2 and HA-tagged βTrCP2. Notably, in co-immunoprecipitation experiments, βTrCP2 interacted to a lower extent with TM-Nrf2 than with WT-Nrf2 (Figure 2A,B), and this impaired interaction became even more apparent in the presence of A-769662, an allosteric AMPK activator (Figure 2C,D). Accordingly, PLA experiments confirmed that Nrf2 interaction with βTrCP2 is impeded upon substitution of the identified serine residues to alanine (Figure 2E). This effect was again pronounced upon AMPK activation by A-769662: whereas WT-Nrf2-transfected cells gave a bright signal, notably predominantly in the nucleus, TM-Nrf2 transfected cells displayed rather low fluorescence (Figure 2F). To exclude experimental artifacts, cells solely expressing either WT-Nrf2, TM-Nrf2, βTrCP2, or pcDNA served as controls. Also, an interaction was exclusively detected between transfected Nrf2 and transfected β TrCP2 since the MYC-Tag of Nrf2 and the HA-Tag of β TrCP2 served as the targets of the PLA primary antibodies. These results so far supported the notion that phosphorylation of S374, S408, and S433 supports Nrf2 degradation by facilitating its interaction with βTrCP2.

### 3.3. Mutation of the AMPK-Dependent Phospho-Sites to Alanine on Nrf2 Impedes Its βTrcP2-Mediated Ubiquitination and Degradation in Keap1^−/−^ Cells

Considering that βTrCP2 is an E3 ubiquitin ligase, we further tested whether its reduced binding to TM-Nrf2 is also reflected by reduced TM-Nrf2 ubiquitination compared to WT-Nrf2. Ubiquitination experiments in KEAP1-depleted MEFs suggested that co-expression of βTrCP2 with wild-type Nrf2, but not with the triple mutated form, resulted in increased ubiquitination of Nrf2 (Figure 3A). Regarding βTrCP2-mediated regulation of Nrf2 protein levels, transient co-transfection of GFP-MYC-WT-Nrf2 with βTrCP2 in MEFs*^Keap1−^*^/*−*^ led to a reduction in the protein levels of WT-Nrf2, and this downregulation was additionally enhanced in the presence of A-769662 (Figure 3B,C). However, mutation of the identified AMPK-dependent phospho-sites in TM-Nrf2 rendered the protein resistant to degradation by co-transfected βTrCP2 and insensitive to AMPK activation (Figure 3D,E). These data corroborate that AMPK-dependent S374, S408, and S433 residues favor the degradation of Nrf2 by facilitating its ubiquitination by βTrCP2.

### 3.4. The Identified AMPK-Dependent Phospho-Sites in Nrf2 Affect the Expression of Endogenous Nrf2 Target Genes

To see whether the aforementioned observations are also of relevance for Nrf2 target gene expression, we examined WT-Nrf2- and TM-Nrf2-mediated transactivation in our model system of MEFs*^Keap1−^*^/*−*^ co-transfected with a βTrCP2 expression plasmid. Specifically, we investigated the transactivation of endogenous *Hmox1* (heme oxygenase 1) and *Gclc* (Glutamate-cysteine ligase catalytic subunit) [71,72] in the presence or absence of A-769662 at two time points. *Hmox1* was expressed to a similar extent by both Nrf2 forms with or without AMPK activation at 4 h (Figure 4C) and 24 h (Figure 4D). Of note, whereas mRNA levels of *Gclc* exhibit similar expression in the presence of either Nrf2 form irrespective of AMPK activation at 4 h (Figure 4A), *Gclc* mRNA was significantly elevated in the case of ΤΜ-Nrf2 in comparison to WT-Nrf2 at 24 h (Figure 4B), suggesting that the more stable TM-Nrf2 also leads to a more sustained *Gclc* expression. Of note, A-769662 elicited elevated mRNA levels of *Gclc* both in the WT- and the TM-Nrf2 setting. This could likely be attributed to other AMPK-dependent signals besides S374, S408, and S433 phosphorylation that can still positively affect the (endogenous) Nrf2-dependent gene transcription in our model, such as phosphorylation of Nrf2 on Ser558, which has been shown to induce nuclear accumulation of Nrf2 [61]. Overall, our findings suggest that AMPK-dependent phosphorylation of S374, S408, and S433 is involved in βTrCP2-mediated control of Nrf2 protein levels and activity, which may gain increased importance in a KEAP1-deficient background.

## 4. Discussion

This study shows that previously identified AMPK-dependent phospho-sites in Nrf2 (S374/408/433) may participate in the regulation of Nrf2 stability by βTrCP2. Mutation of these serine residues to alanine (TM-Nrf2) reduced (i) interaction of Nrf2 with βTrCP2 and (ii) subsequent ubiquitination. Resulting sustained levels of mutated Nrf2 led to (iii) higher expression of the endogenous Nrf2 target gene *Gclc* in *Keap1^−^*^/*−*^ cells (see Figure 5).

In the presence of the A 769662 AMPK activator, our PLA data suggested an enhanced nuclear interaction of both wild-type and mutant Nrf2 forms with β-TrCP2, which could be explained by AMPK directly regulating other phospho-sites influencing the nuclear transport of Nrf2 [61]. One could assume that since the mutant version of Nrf2 cannot undergo β-TrCP2-mediated degradation, the only effect of AMPK on the Nrf2 mutant would be to promote its translocation into the nucleus. However, the net effect of the positive (nucleus translocation) and negative (β-TrCP2-mediated ubiquitination and degradation) influence of AMPK is observed in the case of wild-type Nrf2. These findings could indicate a possible dual role of AMPK in the regulation of Nrf2 in the absence of KEAP1.

With regard to the biological relevance of the aforementioned phospho-residues of Nrf2, we could observe that they affect the transactivating capacity in KEAP1-depleted cells. The expression of *Gclc* was higher in the case of the triple mutated form than in the case of wild-type Nrf2, and the presence of A 769662 enhanced the *Gclc* expression by both forms (compared to the DMSO conditions). These findings indicate once more the dual effect of AMPK activation in Nrf2 regulation and its target genes: higher stability of Nrf2, obtained by the conversion of the identified serine residues to alanine and the inhibition of its degradation, can result in higher transactivation of *Gclc*, while AMPK promotes the nuclear translocation [61] regardless of the AMPK-dependent S374/408/433 residues. Therefore, the final outcome in the expression of *Gclc* [72] is a net effect of the dual regulatory role of AMPK on Nrf2. However, this difference in Gclc gene expression was only observed at a 24 h time point. Thus, it is even tempting to speculate that the sites play a major role in signal termination or duration (via removal of nuclear Nrf2 after target gene expression has started) rather than initiation (nuclear translocation and start of transcription).

Contrary to the *Gclc* results, the analysis of *Hmox1* [71] mRNA expression levels revealed no significant alterations between the Nrf2 forms. A possible explanation of this outcome may lie in the suppressive role of Bach1 in *Hmox1* expression [73]. Downregulation of Bach1 might unmask any possible differences in *Hmox1* gene expression between the Nrf2 forms.

Notably, the same phosphorylation sites that elicited an apparently negative impact on Nrf2 target gene expression in *Keap1^−^*^/*−*^ cells did the opposite in a previous study using cells with a wild-type background, at least for a subset of target genes [65]. This suggests that the sites might act in a dual fashion, namely (a) boosting gene-selective transactivation (by a so far unknown mechanism), but also (b) boosting decay of Nrf2 by β-TrCP2. The latter effect becomes only obvious when Nrf2 is unleashed from the control by its dominant inhibitor, KEAP1.

Degradation of Nrf2 by β-TrCP is known to require preceding priming before phosphorylation by GSK3 in Neh6 to create the necessary phospho-degron, and AMPK activity is known to inhibit GSK3 signaling. Thus, one would assume that activation of AMPK would impede the β-TrCP-mediated destabilization of Nrf2. We, however, propose the acceleration of β-TrCP-conferred degradation via the three AMPK-dependent phospho-sites S374, 408, and 433. A possible resolution of this conflict could lie in the fact that our experimental approaches were conducted in a serum-supplemented medium (10% serum). Under these conditions, GSK3 was already inhibited, and thus GSK3-dependent Neh6 phospho-degron was silenced [74]. It is very likely that we tackled an Nrf2 decay pathway that functions in cells that are depleted from KEAP1 and deprived of GSK3beta activity (as, e.g., possibly found in cancers with KEAP1 mutation and overactive PI3K signaling). One may speculate that the second βTrCP-binding degron in Neh6 (382-DSAPGS-387), which is not phosphorylated by GSK3 [12], could be the one under the influence of the AMPK-dependent phospho-sites S374, 408, and 433 [11]. This hypothesis needs further investigation, e.g., regarding the role of the individual phospho-sites as potential priming or docking sites for βTrCP2. Moreover, it remains to be deciphered why AMPK activation negatively affects the GSK3-dependent 343-DSGIS-347 degron but positively affects the 382-DSAPGS-387 degron, and whether AMPK is truly the one and only kinase directly adding phosphates to serine 374, 408, and 433.

Our study is based on in vitro experiments that use genetically modified mouse embryonic fibroblasts, transiently transfected with fusion proteins. Although this model system allows straightforward and comparably fast investigation of the (biochemical) question of interest, it bears some risks. These include that (i) overexpression of transfected proteins may disrupt endogenous protein stoichiometry, (ii) fusion proteins may behave somewhat differently compared to native proteins, (iii) detection of transfected fusion proteins by antibodies directed against the tag only misses the interference of endogenous counterparts, and (iv) cells in culture cannot recapitulate the situation in vivo (e.g., cell–cell competition for fuel, oxygen tension, and organ–organ interaction). With this in mind, we still put forward a novel addition on how AMPK signaling could intersect with the Nrf2-mediated stress response, namely by triggering faster Nrf2 destabilization via phosphorylation, facilitated βTrCP2 binding, and ubiquitination, for further investigation in advanced test models. Our data further support the notion that AMPK activation with its pleiotropic effects can have a positive or negative impact on NRF2 signaling: on the one hand, it can boost Nrf2 signaling (e.g., by increasing nuclear accumulation or autophagic degradation of KEAP1) and on the other hand, it can also dampen the Nrf2 signal (e.g., by accelerated degradation). To which side the balance tips may highly depend on cellular environment, genetic background, Nrf2 target gene, signal hierarchy, and integration. Our findings shed light on the regulatory mechanisms governing Nrf2 and provide new insights into the multifaceted roles of AMPK in cellular signaling. A better understanding of the intricate interplay between AMPK and NRF2 signaling with all possible parameters by future studies, including in vivo and bioinformatics approaches, will hopefully allow safe and reliable exploitation of pharmacological AMPK activation in conditions of aberrant Nrf2 signaling.

## 5. Conclusions

In this study, we elucidated the role of the AMPK-dependent phosphorylation sites (S374, S408, and S433) of Nrf2 in its βTrCP2-driven degradation in a KEAP1-depleted background. Mutation of the three serines to alanine impeded the interaction of Nrf2 with βTrCP2 and its βTrCP2-mediated ubiquitination and degradation. Consequently, higher stability of Nrf2 positively affected gene expression of Nrf2 targets. Thus, modulating the AMPK-Nrf2 crosstalk could decrease cellular stress resilience via diminished Nrf2 stability, which would be of use in the context of diseases with overactive Nrf2 signaling. This hypothesis warrants further investigation in appropriate model systems.

## Figures and Tables

**Figure 1 antioxidants-12-01586-f001:**
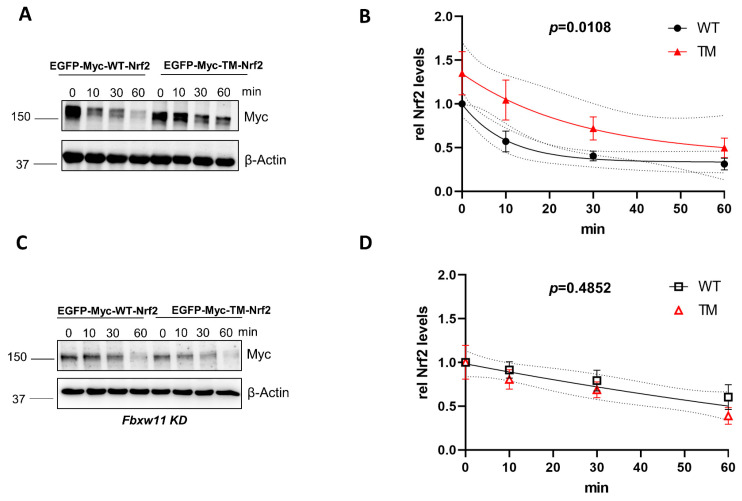
Mutation of the AMPK-dependent phospho-sites to alanine stabilizes Nrf2 in *Keap1^−^*^/*−*^ cells in a βTrCP2-dependent manner. (**A**,**B**). *Keap1*^−/−^ MEFs were transfected with EGFP-MYC tagged WT- or TM-Nrf2 expression plasmids for 48 h and then exposed to cycloheximide (50 μM) for different periods of time. Cell lysates were subjected to immunoblot analysis for MYC or β-Actin. A representative blot (**A**) and quantification analysis (**B**) of n = 9 independent experiments (relative to signal at t = 0) are depicted. One-phase exponential decay curves were fitted to the data by finding the least sum of squares using GraphPad Prism. The curves were significantly different, with a *p* value = 0.0108. (**C**,**D**) *Keap1^−^*^/*−*^ MEFs were transfected with *Fbxw11*-specific siRNA and with EGFP-MYC tagged WT- or TM-Nrf2 expression plasmids for 48 h, and then exposed to cycloheximide (50 μM) for different periods of time. Cell lysates were subjected to immunoblot analysis for MYC or β-Actin. A representative blot (**C**) and quantification analysis (**D**) of n = 5 independent experiments (relative to signal at t = 0) are depicted. One-phase exponential decay curves were fitted to the data by finding the least sum of squares using GraphPad Prism. There was one curve for all data sets with a *p* value = 0.4852. The dotted lines in (**B**,**D**) represent the confidence bands with a 95% confidence level. Data are presented as means  ±  SEM.

**Figure 2 antioxidants-12-01586-f002:**
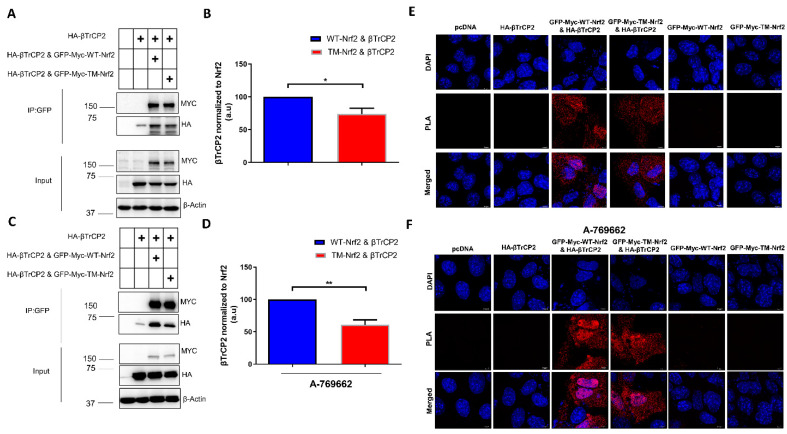
Mutation of the AMPK-dependent phospho-sites to alanine impedes the interaction of Nrf2 with β-TrCP2 in *Keap1^−^*^/*−*^ cells. (**A**–**D**) *Keap1*-null MEFs were transfected with an expression plasmid for HA-tagged β-TrCP2 alone or together with constructs encoding EGFP-MYC-WT-Nrf2, the triple mutated version EGFP-MYC-TM-Nrf2, or pcDNA as indicated for 48 h followed by the absence (**A**,**B**) or presence (**C**,**D**) of the AMPK activator A769662 (50 μM, 4 h). Cells were also treated at the same time with MG132 (10 μM, 4 h) to stabilize Nrf2, which was pulled down using GFP-Trap. Precipitated proteins, as well as input controls, were immunoblotted for MYC, HA, and β-Actin. Representative blots (**A**,**C**) and quantification analysis (**B**,**D**) of 3 independent experiments are depicted. Plus symbols in the tables indicate the different constructs/chemicals used in the various experimental conditions. (**E**,**F**) *Keap1*-null MEFs were transfected with (i) pcDNA, (ii) HA-tagged β-TrCP2, or co-transfected with (iii) HA-tagged β-TrCP2 and EGFP-MYC-tagged WT-Nrf2, (iv) HA-tagged β-TrCP2 and EGFP-MYC-tagged TM-Nrf2 expression plasmids, or (v) EGFP-MYC-tagged WT-Nrf2 and (vi) and EGFP-MYC-tagged TM-Nrf2 alone, and then treated with MG132 (10 μM) for 4 h in the absence (**E**) or presence (**F**) of the AMPK activator A769662 (50 μM, 4 h). Cells were fixed with 4% Paraformaldehyde and subjected to PLA. Data are presented as means ± SEM. Statistical analysis was performed using Student’s *t*-test, * *p* < 0.05 and ** *p* < 0.01.

**Figure 3 antioxidants-12-01586-f003:**
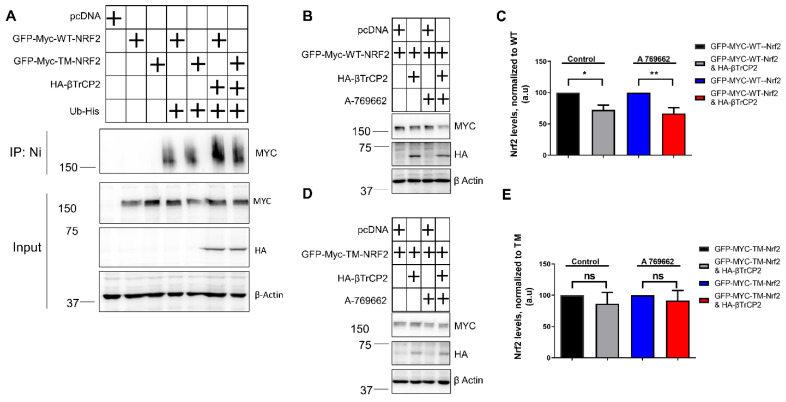
Mutation of the AMPK-dependent phospho-sites to alanine on Nrf2 impedes its βTrcP2-mediated ubiquitination and degradation in *Keap1^−^*^/*−*^
*cells*. (**A**–**E**) *Keap1*-null MEFs were transfected with the indicated expression plasmids for 48 h. (**A**) Ubiquitination assay of EGFP-Myc tagged Nrf2 expression plasmids in *Keap1*-null MEFs, after co-transfection with His-tagged Ubiquitin and the indicated HA-βTrCP2 construct. (**B**–**E**) Expression levels of EGFP-Myc tagged WT-Nrf2 (**B**,**C**) or EGFP-Myc tagged TM-Nrf2 (**D**–**E**) with or without co-transfection of HA-tagged βΤrCP2 in the presence or absence of the AMPK activator A 769662 (50 μM, 4 h). Cell lysates were subjected to immunoblot analysis for MYC, HA, or actin. Representative blots (**B**,**D**) and quantification analysis (**C**,**E**) of 3 independent experiments are depicted. Plus symbols in the tables indicate the different constructs/chemicals used in the various experimental conditions. Data are presented as means ± SEM. Statistical analysis was performed using multiple comparison one-way ANOVA, with Sidak’s correction, * *p* < 0.05 and ** *p* < 0.01. ns: not significant.

**Figure 4 antioxidants-12-01586-f004:**
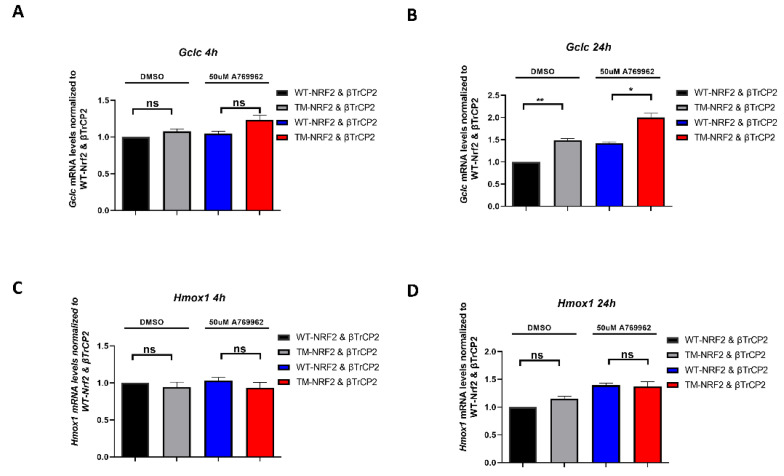
The identified AMPK-dependent phospho-sites in Nrf2 affect the expression of endogenous Nrf2 target genes. (**A**–**D**) *Keap1*-null MEFs were transfected for 48 h with an expression plasmid for HA-tagged β-TrCP2 together with constructs encoding EGFP-MYC-WT-Nrf2 or the triple mutated version EGFP-MYC-TM-Nrf2, as indicated in the absence or presence of the AMPK activator A769662 (50 μM, 4 h and 24 h). RNA was extracted and analyzed for the abundance of *Gclc* (**A**,**B**) and *Hmox1* (**C**,**D**) by qPCR (TBP as reference gene). Data are presented as means ± SEM. Statistical analysis was performed using multiple comparison one-way ANOVA, with Sidak’s correction, * *p* < 0.05 and ** *p* < 0.01. ns: not significant.

**Figure 5 antioxidants-12-01586-f005:**
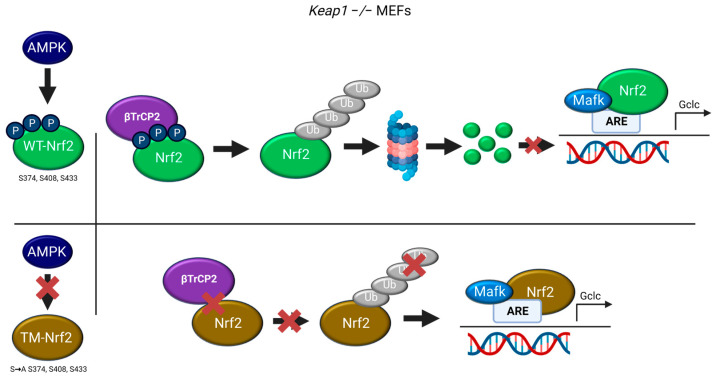
Working model depicting the role of the AMPK-dependent S374, S408, and S433 of Nrf2 in its βTrCP2-mediated ubiquitination and degradation, regulating the expression of Nrf2 target genes (©BioRender).

## Data Availability

The data presented in this study are available upon request from the corresponding author.

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
