# Peer review of "AMPK-Mediated Phosphorylation of Nrf2 at S374/S408/S433 Favors Its βTrCP2-Mediated Degradation in KEAP1-Deficient Cells"

_antioxidants, 2023, doi:10.3390/antiox12081586_

Round 1
Reviewer 1 Report
The authors previously identified three AMPK-dependent phosphorylation sites (S374/408/433) in Nrf2. This study examined whether these AMPK-dependent phosphorylation sites may influence the βTrCP-driven degradation of Nrf2. Their results show that TM-Nrf2S→A374/408/433 had enhanced stability due to impeded interaction with βTrCP2 and reduced ubiquitination of Nrf2. The authors concluded that targeting AMPK might be able to diminish Nrf2-mediated responses in cells with overactive Nrf2 due to KEAP1 deficiency. This interesting original article clarified the crosstalk between NRF2 and AMPK. The overall structure is of good quality, the methods and results are very clear, figures are of good quality and clear to readers. However, some controls are missing. Comments are listed below:
1. Please explain in the method why not to directly use Nrf2 antibody to detect the protein expression of Nrf2.
2. It will be better to provide some results that demonstrate the endogenous protein levels of Keap1, Nrf2, and βTrCP2 in Keap1-null MEFs. Does the endogenous Nrf2 interfere with these observations?
3. In the Figure 1 legend, “Keap1‐/‐ MEFs were transfected with Fbxw11-specific sgRNA (sgFbxw11).” Is sgRNA an error of siRNA?
4. Dose AMPK activator A769662 increase the phosphorylation of Nrf2 at S374, S408, S433 residues? These data should be provided.
5. There is no direct evidence that Nrf2 was translocated to cell nuclei. Nuclear Nrf2 staining with Nrf2 antibody should be provided.
6. The role of AMPK was not indicated in Figure 5.
Reviewer 2 Report
The manuscript is almost well written.
Overall the topic could be interesting but some details could be improved.
I recommend that the paper be accepted with minor revision:
The authors should better explain the purpose of the work
It is possible to translate the work in vivo, there are other similar jobs
The authors should add a conclusion paragraph
Please correct spelling and grammatical errors
Reviewer 3 Report
The authors examine the function of AMPK in Nrf2 phosphorylation and further degradation pathways in genetically modified mouse fibroblast cells. They locate specific binding sites on Nrf2 for inducing proteasomal degradation, other than the KEAP1-regulated pathway. Besides, they prove the regulatory role of AMP in the above process, showing a relationship between the metabolic and energetical status of the cell and its reaction to oxidative and other stresses.
The experiments are well-planned, and the methods used are modern and elegant. The findings are new and well-characterized. The consequences of the results are correct, also the explanation of the possible deficiencies of these types of in vitro studies. English is flawless; the paper is easy to read and understand. All my possible questions are answered in advance. I accept the article in its present form.
Round 2
Reviewer 1 Report
The authors have adequately responded to the comments. The error " Fbxw11-specific sgRNA" is still not corrected to siRNA in Figure 1 legend.